# PROBELM: Plausibility Ranking Evaluation for Language Models

**Zhangdie Yuan, Eric Chamoun[†], Rami Aly[†], Chenxi Whitehouse[†], Andreas Vlachos**
Department of Computer Science and Technology
University of Cambridge
{zy317,ec806,rmya2,av308}@cam.ac.uk, chenxi.whitehouse@cl.cam.ac.uk

## Abstract

This paper introduces PROBELM (Plausibility Ranking Evaluation for Language Models), a benchmark designed to assess language models' ability to discern more plausible from less plausible scenarios through their parametric knowledge. While benchmarks such as TruthfulQA emphasise factual accuracy or truthfulness, and others such as COPA explore plausible scenarios without explicitly incorporating world knowledge, PROBELM seeks to bridge this gap by evaluating models' capabilities to prioritise plausible scenarios that leverage world knowledge over less plausible alternatives. This design allows us to assess the potential of language models for downstream use cases such as literature-based discovery where the focus is on identifying information that is likely but not yet known. Our benchmark is constructed from a dataset curated from Wikidata edit histories, tailored to align the temporal bounds of the training data for the evaluated models. PROBELM facilitates the evaluation of language models across multiple prompting types, including statement, text completion, and question-answering. Experiments with 10 models of various sizes and architectures on the relationship between model scales, training recency, and plausibility performance, reveal that factual accuracy does not directly correlate with plausibility performance and that up-to-date training data enhances plausibility assessment across different model architectures. [1]

## 1 Introduction

Recent advancements in large language models (LLMs) have significantly enhanced their performance across a wide range of NLP tasks. Alongside these developments, various benchmarks and datasets are introduced to effectively assess the capabilities of LLMs, particularly in terms of knowledge and reasoning (Roemmele et al., 2011; Wang et al., 2018; Clark et al., 2018; Zellers et al., 2019; Wang et al., 2019; Sakaguchi et al., 2021; Lin et al., 2022; Whitehouse et al., 2023; Li et al., 2023). However, these benchmarks often focus predominantly on evaluating factual accuracy or reasoning abilities without explicitly incorporating broader world knowledge.

For instance, benchmarks like TruthfulQA (Lin et al., 2022) are designed to assess the truthfulness or factual correctness of LLMs, evaluating their ability to retrieve and apply information encoded during training, including tasks like mathematical induction. However, they do not explicitly address LLMs' capacity to discern plausibility in scenarios where strict factual accuracy might not be directly applicable. On the other hand, datasets such as COPA (Choice of Plausible Alternatives) (Roemmele et al., 2011) evaluate models through causal reasoning tasks, asking the model to choose the more plausible scenarios from the two options for a given premise. For example, given a premise "I tipped the bottle", scenario "The liquid in the bottle poured out" is more plausible than "The liquid in the bottle froze". While COPA extends evaluation beyond mere factual accuracy by introducing

---

[†]Equal Contribution. More senior authors are listed towards the end of the author list.
[1]Our dataset and code are available at https://github.com/zhangdiey/PRobELM

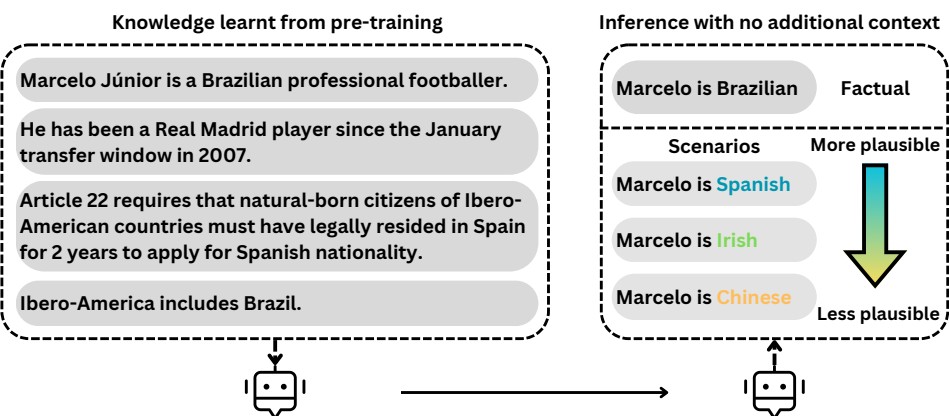

Figure 1: This figure illustrates the utility of plausibility evaluation in language model outputs: a model, trained on data up until 25 July 2011, ranks "Marcelo is Spanish" as the most plausible among non-factual scenarios. This judgement is based on the model's parametric knowledge of Marcelo's career with a Spanish club, his residency in Spain and the Spanish naturalisation law. The plausibility of this scenario is confirmed as Marcelo was granted Spanish citizenship on 26 July 2011.

plausible alternative scenarios, it operates within the constraints of an artificially constructed dataset, potentially oversimplifying evaluation by limiting tasks to binary choices. Notably, fine-tuned state-of-the-art models such as PaLM and PaLM 2 have shown near-perfect performance on COPA (Chowdhery et al., 2023; Anil et al., 2023), indicating a necessity for more complex and challenging benchmarks that better capture the subtleties of world knowledge and plausibility.

Furthermore, in certain domains, combining broad world knowledge and assessing plausibility is crucial. One such domain is literature-based knowledge discovery, particularly within the biomedical field, which can be accelerated by guiding experimental efforts leveraging existing literature (Gopalakrishnan et al., 2019), requiring the capability of deriving educated guesses that may not be immediately verifiable. In this context, the ability of LLMs to navigate through and infer plausible scenarios from world knowledge becomes invaluable. Yet, as SCIMON (Wang et al., 2023) demonstrates, while LLMs can generate hypotheses, they often lack technical depth and novelty (Wang et al., 2023), hindering their contribution to specialised fields.

To address the limitation in plausibility evaluation, we introduce PROBELM, a novel benchmark designed to directly assess the plausibility inference capabilities of LLMs. PROBELM utilises scenarios collected from real-world data through Wikidata, comprising two primary components: new facts sourced from Wikidata that were unknown to models due to the timeframe of their training data, and a set of automatically generated less plausible scenarios, and asks models to rank scenarios from most to least plausible. For instance, consider a scenario where a model is asked to determine a person's (Marcelo) nationality based on provided facts about their football career and immigration laws, as illustrated in Figure 1. Provided with the available information in the training data of a model such as "Marcelo is a Brazilian footballer" and "He has played for a Spanish club since 2007", along with pertinent legal information regarding immigration, like "Ibero-American natural-born citizens must have lived in Spain for 2 years to apply for citizenship" and "Ibero-America includes Brazil", the model is expected to rank "Spanish" as the most plausible nationality, reflecting its ability to reason with encoded parametric knowledge.

We evaluate PROBELM with 10 language models varying in architecture, parameter scale, and the recency of their world knowledge updates. To ensure alignment in the pre-training data sourced from Wikipedia, we select open-source models that disclose the timestamp of the utilised Wikipedia dump in their training, including GPT-2 (Radford et al., 2019), BLOOM (Scao et al., 2023), LLaMA (Touvron et al., 2023), Pythia (Biderman et al., 2023), and

OLMo (Groeneveld et al., 2024), spanning parameter counts from 14 million to 7 billion and training data from 2017 to 2023. Our empirical results reveal the following key observations:

- Models with higher performance on factual accuracy do not necessarily excel in plausibility, indicating a divergence between factuality and plausibility capabilities.
- Larger models generally exhibit better plausibility inference, but this trend varies across model families.
- Models trained on less recent data can sometimes outperform counterparts that use more contemporary datasets, demonstrating that model architecture and training methodologies also influence plausibility inference, independent of model size.
- A notable trend suggests that the greater the temporal gap between a model's training data cut-off and the date of the evaluation set, the poorer the model performs on PROBELM, underscoring the importance of up-to-date knowledge in evaluating plausibility.

## 2 Related Work

With the rise of more powerful language models in the form of LLMs, sophisticated reasoning benchmarks such as the Open LLM Leaderboard have been proposed. The benchmark incorporates tasks such as HellaSwag (Zellers et al., 2019), which evaluates commonsense reasoning within physically situated contexts. Similarly, COPA confronts models with commonsense scenarios. Yet, these benchmarks do not task the model to incorporate knowledge about the world state into its prediction. On the flip side, benchmarks such as TruthfulQA (Lin et al., 2022), FactScore (Min et al., 2023), and HaluEval (Li et al., 2023; Yu et al., 2023) measure the factuality of a model's response exclusively. However, previous benchmarks rarely address the complex *interplay* between world knowledge and plausibility.

Factual errors in an LLM's output are also referred to as hallucinations (Xu et al., 2024) and occur when models generate content that is unfaithful to the provided input, established context, or factual correctness. Ji et al. (2023) categorises these into *input-conflicting*, *context-conflicting*, and *fact-conflicting* hallucinations, with the latter category being particularly problematic due to its contradiction with known facts.

While all scenarios in the PROBELM benchmark are technically fact-conflicting hallucinations (as they are not yet facts of the world the LLM has seen), the crucial point is that under existing world knowledge, non-factual scenarios are not all equally plausible. Instead, PROBELM probes a model's ability to make educated guesses about plausible futures to gain insights into the capabilities of LLMs to infer within the limits of their knowledge beyond recalling facts it has already seen.

The importance of a benchmark that measures the interplay between world knowledge and plausibility in the context of hallucinations is underscored by recent work. Hao et al. (2023) repurpose LLMs as both a world model and a reasoning agent for planning while Gruver et al. (2023) show that LLMs are time series forecasters, generating plausible continuations for existing data. Despite these positive signals, Mündler et al. (2024) observes a significant presence of self-contradictory outputs in LLMs, concretely in 17.7% of all sentences produced by ChatGPT, indicating inconsistent use of world knowledge. This finding is supported by Ye et al. (2024), who observe that larger-scale LLMs demonstrate greater uncertainty compared to their smaller counterparts. Finally Kim et al. (2023), highlighted LLMs' difficulties in maintaining consistent reasoning through the FANToM framework.

## 3 PROBELM Construction

PROBELM is inspired by acknowledging the middle ground between absolute fact and pure fiction, a domain abundant with feasible scenarios. These scenarios, grounded in logic yet not strictly confined to facts, are crucial for applications such as knowledge discovery. Plausible inferences are valuable in exploring existing literature and uncovering potential connections or hypotheses that may spark meaningful scientific investigation, even if not

immediately verifiable. PROBELM seeks to address a significant gap in contemporary evaluation paradigms, establishing a novel benchmark for assessing the proficiency of LLMs in navigating the complexities of real-world knowledge application.

In the illustrative scenario provided in Figure 1, a model trained with data available until the 25th of July, 2011, is asked to evaluate the plausibility of various nationalities for the Brazilian footballer, Marcelo. This task requires the model to engage in deductive reasoning, utilising its accumulated knowledge to prioritise potential nationalities based on their likelihood. Specifically, the model is expected to integrate the information pertaining to Marcelo's eligibility for Spanish citizenship, thereby identifying "Spanish" as a plausible nationality. This example highlights the PROBELM's focus on models' ability to perform deductive reasoning and to apply world knowledge.

PROBELM assesses language models' capacity to determine plausibility through three different structured prompts: statements, text completions, and direct inquiries or question answering (see Table 8 in Appendix D for examples).

**Statements:** Models are presented with declarative sentences that depict different scenarios such as "Marcelo is [*possible scenario*]", where *possible scenario* could be indicative of nationalities, for instance, Spanish, Brazilian, or Chinese. The model is asked to calculate perplexity scores for each sentence, enabling the ranking of statements based on plausibility, with lower perplexity scores indicating higher likelihood.

**Text Completion:** In this prompt format, models are given sentences structured to require completion, such as "Fill in the blank: Marcelo has citizenship of ___. Answer: [*possible scenario*]"'. The models are then tasked with calculating perplexity scores for different *possible scenarios*, ranking them from most to least plausible based on perplexity.

**Question Answering:** Here, models are presented with direct questions, such as "What citizenship(s) does Marcelo have? Answer: [*possible scenario*]." where a *possible scenario* could be Spanish, Irish, or Chinese, similarly to the previous examples. The model evaluates the perplexity of each question-answer pair, arranging answers by plausibility according to calculated perplexities.

The primary aim across these varied formats is not to generate novel text but to compare perplexity scores for the given scenarios. This methodology seeks to measure a model's proficiency in identifying the most plausible scenario among alternatives, leveraging the context embedded within each prompt type. Such an approach is crucial for assessing a model's capability to evaluate the likelihood or credibility of different scenarios based on the specific context.

### 3.1 Most Plausible Scenarios

Evaluating language models on PROBELM is challenging due to its subjective nature and susceptibility to biases, including ethical concerns. To address these complexities, our methodology incorporates a novel strategy grounded in probabilistic reasoning, with an emphasis on Bayesian inference principles. Bayesian inference posits that the likelihood of a hypothesis evolves in response to new evidence. Accordingly, our approach assumes that the most immediate future events, as sequentially recorded in successive updates of Wikidata, represent the most plausible scenarios compared to our current knowledge base. This assumption rests on the motion that the near future is inherently more predictable and grounded in the current state of the world, thereby providing a robust benchmark for evaluating the plausibility of language model predictions.

In practice, the first step of our methodology is to determine the recency of each language model's training data, e.g., OLMo's dataset extends up to March 2023. Subsequently, we identify the most plausible scenarios by analysing Wikidata revisions immediately following this cutoff (e.g., in OLMo's case, between March 2023 and the subsequent update).[2] This approach aims to capture recent facts and developments that were not part of the model's knowledge during training. By focusing on Wikidata changes after the training dataset's

---

[2]We employ TemporalWiki (Jang et al., 2022) to generate the edits between two Wikidata dumps.

timeframe, we pinpoint instances that, though are *now* confirmed facts or events, represent highly plausible scenarios to the models due to their temporal alignment with the immediate future. This method ensures the scenarios selected for evaluation are both relevant and challenging, providing a comprehensive test of the models' capacity to utilise the world knowledge available at the time of their last update. Consequently, this creates a dynamic and adaptive benchmark that accesses the models' plausibility inference abilities.

## 3.2 Less Plausible Scenarios

Within the PROBELM framework, the construction of less plausible scenarios leverages a statistical technique grounded in examining entity co-occurrences within Wikidata. By systematically altering the entities and their attributes in a given plausible scenario—while maintaining the relationship unchanged—we assess the plausibility of these variations according to their occurrence frequency throughout the dataset. This assessment extends to a wide spectrum of entity associations, beyond isolated instances, ensuring a comprehensive evaluation of plausibility across varied scenarios.

Below is an example of how we assess the plausibility of different scenarios using Wikidata co-occurrences. Take the case of Marcelo, who is known to be Brazilian, and evaluate the plausibility of the triple <Marcelo, has citizenship, Spanish>. We assess the likelihood of the "Brazilian" and "Spanish" attributes being linked under the "has citizenship" relation by calculating their co-occurrence within Wikidata when Marcelo is replaced with different entities. Given the rules facilitating Spanish naturalisation for Brazilians (Figure 1), these attributes are likely to co-occur much more frequently than "Brazilian" with "Chinese" for instance, due to China's policy disallowing dual citizenship. Thus, the plausibility of "Chinese" citizenship in this context is diminished. These co-occurrence scores facilitate the construction of a plausibility hierarchy based on statistical evidence from real-world data.

By maintaining the logical structure of relations such as "has citizenship" and manipulating the entities and their associated attributes, we generate scenarios from plausible to less plausible. Our approach not only leverages the rich relational data in Wikidata but also mirrors the complexity of real-world knowledge and its application, providing a robust framework for the plausibility assessment of language model outputs. The detailed algorithm for generating these scenarios, including the consideration of entities with complex attributes like dual citizenship, is elaborated in Appendix B.

## 3.3 Quality Control

To ensure the generated scenarios are both accurate and relevant, we implement several quality control measures to address potential biases and inaccuracies in the dataset:

**Paraphrase Discrimination**: To address the presence of repetitive entities in Wikidata, we employ FuzzyWuzzy[3], a tool that calculates sequence differences using the Levenshtein Distance. This method helps in identifying and filtering out paraphrases, thereby reducing redundancy and ensuring the dataset reflects a diverse range of entities. By distinguishing between similar yet distinct entities, FuzzyWuzzy enhances the precision of our dataset.

**Manual Filtering of Non-event Edits**: We manually review Wikidata edits to exclude those not representing event changes. This involves creating a whitelist of relations representing events, such as "place of death", ensuring the dataset emphasises eventful modifications, and improving the relevance of our scenarios for plausibility assessment. The full whitelist of relations is presented in Appendix D.

**Manual Filtering of General Queries**: To avoid over-generic scenarios, we manually filter out general queries derived from the most plausible triples, prioritising queries that yield more specific and insightful scenarios. In the example of <Marcelo, has citizenship, Spanish>, we consider the query on "Which citizenship does Marcelo have?" to be more specific than "Who else is a Spanish citizen?".

---

[3]https://pypi.org/project/fuzzywuzzy/

| Model (size) | Pre-training Data Up To | Evaluation Timestamps | Number of Scenarios |
|---|---|---|---|
| GPT-2 (124M, 1.5B) | December 2017 | Dec 2017 - Jan 2018 | 9,328 |
| BLOOM (560M, 7B) | December 2021 | Dec 2021 - Jan 2022 | 8,569 |
| LLaMA (7B) | August 2022 | Nov 2023 - Dec 2023 | 5,280 |
| Pythia (14M, 160M, 2.8B) | March 2020 | Oct 2020 - Nov 2020 | 5,104 |
| OLMo (1B, 7B) | March 2023 | Nov 2023 - Dec 2023 | 5,280 |

Table 1: Evaluated models and their corresponding data timelines in the PROBELM benchmark. This table highlights the diversity of model sizes, their training data recency, and the specific timestamps chosen for evaluating their plausibility inference capabilities.

**Template Design for Natural Language Conversion**: We manually create templates for each relation to convert triples into consistent and natural sentences. This is essential for preserving scenario coherence and readability, facilitating accurate plausibility inference by language models. Customising templates for each relation ensures grammatical correctness and contextual appropriateness, thus improving the dataset's interpretability. The list of the templates used is also illustrated in Appendix D.

## 3.4 Comparison with Other Datasets

Our dataset demands leveraging extensive world knowledge, making the task more complex and relevant for knowledge discovery compared to datasets like COPA which focus on commonsense reasoning without extensive world knowledge. For example, the scenario "I tipped a bottle" leading to "The liquid poured out" is self-contained. In contrast, scenarios like the Brazilian footballer Marcelo being plausibly Spanish or Portuguese require extensive background knowledge. Our dataset acknowledges the existence of multiple plausible events for a given scenario and includes ten less plausible scenarios to ensure a challenging evaluation environment, reducing the likelihood of models performing well by random guessing. Negative sampling is designed around event relations, with all 71 relations and 213 templates detailed in Appendix D, ensuring a diverse and representative set of scenarios.

Also, note that our concept of plausibility extends beyond identifying surprising events. It targets scenarios that, while not present in the training data of language models LLMs, have subsequently become facts. For instance, the proposition "the Earth is round" would have been highly plausible based on existing knowledge, even though it was not widely accepted in ancient times. This highlights the importance of plausibility in knowledge discovery, distinguishing it from predicting future events. Additionally, while individual timestamps in Wikidata might suggest a Markovian approach, our model does not strictly adhere to this. The world knowledge and historical events encoded in LLMs are inherently non-Markovian, providing a richer context for evaluating plausibility. This broader perspective is crucial for practical and comprehensive plausibility assessments.

As the dataset is timeframe-specific regarding the models used for evaluation, we include the statistical details in the following section.

## 4 PROBELM as a Plausibility Benchmark

We evaluate PROBELM on 10 models with different sizes and architecture, and the timeframe of training. To ensure alignment in the pre-training data sourced from Wikipedia, we select open-source models that disclose the timestamp of the utilised Wikipedia dump in their training, resulting in models including GPT-2, BLOOM, LLaMA, Pythia, and OLMo. In the following, we detail our experimental setup and the evaluation metrics.

### 4.1 Experimental Setup

We first collect data points from Wikidata that align the timeframe of a given model (outlined in Table 1), following the steps detailed in section 3. For each most plausible scenario, we

always add 10 less plausible alternatives, resulting in a total of 11 scenarios for models to rank. To ensure our evaluation is comprehensive yet with the practical constraints of time and computational resources, we sampled 126,000 scenarios (6,000 most plausible scenarios and 120,000 less plausible scenarios) from each timestamp, leading to four distinct timeframe-specific evaluation datasets. This sampling covered the five models under study, with OLMo and LLaMA models sharing the same timestamp. Details of the models and the corresponding timestamps are included in Appendix A.

Next, we follow the quality control measurements as detailed in subsection 3.3 to filter our data points. The final number of samples for each timestamp varies, as shown in Table 1.

For each set of one most plausible scenario alongside its 10 less plausible alternatives, we probe the models using the three types of prompts described in section 3, employing a zero-shot approach. We calculate the perplexity for these 11 scenarios for each prompt type and then rank them based on their perplexity[4] scores.

Note that PROBELM is designed to capture unknown or unrecorded events that can be inferred from existing world knowledge, emphasizing knowledge discovery over sequential event forecasting. Effective models should synthesize knowledge plausibly, as illustrated by the example of Marcelo, where evidence should favor plausible over implausible nationalities. Our benchmark remains relevant as long as the training data of LLMs predate the evaluation timestamps. For instance, both OLMO and LLama were evaluated using the same benchmark version, with timestamps more recent than their training data. The benchmark construction process is automated, allowing for easy updates with more recent data if needed, ensuring continued relevance for future models.

## 4.2  Evaluation Metrics

In assessing the performance of language models on the PROBELM benchmark, we utilise a set of evaluation metrics designed to capture various aspects of ranking plausibility. Each metric offers a different perspective on the models' ability to discern and prioritise plausible scenarios, providing a comprehensive overview of their capabilities.

**Accuracy** focuses on the the top-ranked scenario. This metric determines whether the model can identify the most plausible scenario as its first choice. However, accuracy does not account for the placement of scenarios beyond the top-ranked ones, making it a measure of immediate accuracy rather than a reflection of the overall ranking quality. We note that *accuracy* here is different from the factual correctness, as none of the ranked scenarios is factual but ranges from most to least plausible.

**Mean Reciprocal Rank** (MRR) is employed to calculate the average of the reciprocal ranks of the most plausible scenario across all queries. MRR offers insight into the model's ability to rank the most plausible scenario highly but does not evaluate the arrangement of other scenarios in the list, emphasising the importance of the top plausible scenario's position within the model's ranking.

**Normalised Discounted Cumulative Gain** (NDCG) is utilised to assess the quality of the entire ranked list produced by the model. NDCG takes into account the graded relevance of all ranked scenarios, providing a measure of the list's overall quality and the effectiveness of the model's ranking across different levels of plausibility. Unlike accuracy and MRR, NDCG does not isolate its evaluation to any single scenario, such as the top-ranked or the most plausible one, but instead evaluates the cumulative relevance of the whole list, rewarding rankings that place more plausible scenarios higher.

In addition, we measure the PLAUSIBILITY score, where we average accuracy, MRR, and NDCG scores across the three different prompt types.

---

[4]From a certain perspective, beam search could also be used to obtain more plausible hypotheses, and its use is orthogonal to evaluation with our benchmark. However, maximum a posteriori estimates are not always the best way to decode a model, which is why minimum Bayes risk decoding is popular in fields like machine translation. Our method focuses on evaluating the plausibility of scenarios using perplexity, providing a different perspective than beam search.

| MODEL | AVG | Statement | | | Text Completion | | | Question Answering | | |
|---|---|---|---|---|---|---|---|---|---|---|
| | PLAUSIBILITY | ACC. | MRR | NDCG | ACC. | MRR | NDCG | ACC. | MRR | NDCG |
| OLMO-1B | 45.07 | 29.96 | 49.92 | 50.13 | 22.11 | 46.47 | 46.95 | 42.15 | 59.09 | 58.89 |
| OLMO-7B | 52.51 | 31.20 | 51.62 | 51.57 | **44.62** | 61.87 | 61.43 | 45.45 | 62.60 | 62.23 |
| BLOOM-560M | 30.63 | 20.58 | 43.37 | 42.56 | 8.78 | 34.81 | 34.54 | 12.54 | 39.44 | 39.02 |
| BLOOM-7B | 40.74 | 27.35 | 49.90 | 49.50 | 17.44 | 44.21 | 43.39 | 12.80 | 41.11 | 40.29 |
| GPT-2-124M | 39.02 | 29.08 | 49.08 | 47.96 | 18.09 | 40.84 | 39.83 | 26.97 | 50.12 | 49.19 |
| GPT-2-1.5B | 45.51 | 29.19 | 49.81 | 48.83 | 25.52 | 48.92 | 48.11 | 41.95 | 58.92 | 58.34 |
| LLAMA-7B | 40.74 | 19.20 | 43.26 | 43.60 | 29.96 | 52.08 | 51.81 | 27.69 | 49.56 | 49.51 |
| PYTHIA-14M | 35.90 | 20.68 | 43.15 | 43.11 | 17.67 | 39.48 | 39.37 | 23.90 | 47.90 | 47.85 |
| PYTHIA-160M | 42.45 | 34.54 | 54.39 | 55.03 | 16.87 | 39.59 | 40.13 | 30.52 | 55.16 | 55.84 |
| PYTHIA-2.8B | **58.47** | **43.78** | **61.17** | **61.14** | 43.57 | **62.77** | **62.33** | **53.41** | **69.13** | **68.89** |

Table 2: Evaluation results of language models on PROBELM with corresponding time-frame specific datasets. We report accuracy, MRR, and NDCG scores for three prompt types. AVG indicates average *plausibility* across all prompts and metrics (average of nine scores). Best performances are in bold, and second-best results are underlined.

## 5    Results and Discussion

### 5.1    Main Results on PROBELM

We present the plausibility score, and accuracy, MRR, and NDCG scores of different prompt types in Table 2, using the timeframe-specific datasets for each model.

The results reveal that while model performance exceeded random chance, overall scores remain low, underscoring the complexity of the plausibility assessment task. For instance, considering a random baseline accuracy of 9.09% when ranking 11 scenarios, most models surpass this baseline across various sizes and prompts. Notably, both BLOOM models marginally outperform the random baseline by approximately 3 points, achieving 12.54% and 12.80% on the question-answering prompt. Specifically, the BLOOM-560M model underperform relative to the random baseline in the text completion prompt with 8.78% accuracy. This performance pattern suggests that language models face significant challenges in differentiating between scenarios of varying plausibility. It also highlights a distinct separation between models' capabilities in factual accuracy and their ability to infer plausibility. These observations set the stage for our key findings:

**Larger models generally outperform smaller ones in plausibility tasks, but performance varies across model families.** For example, the ACC. scores across models of varying sizes show 45.07 vs 52.51 for OLMo, 30.63 vs 40.73 for BLOOM, 39.02 vs 45.51 for GPT, and a notable increase from 35.90 through 42.45 to 58.47 for Pythia models. However, this effect varies across different model families. Remarkably, the Pythia 2.8B model outperformed all three 7B models by at least 6 percentage points, while the Pythia 160M model surpassed both LLaMA 7B and BLOOM 7B models—significantly larger in scale—by 2 percent.

These exceptions underscore that the relationship between model size and plausibility assessment is complex and not linear. Performance variability across different model families underscores the profound impact of architectural differences and training methodologies on a model's plausibility inference capabilities, suggesting that these factors are as crucial as, if not more so than, model size. This complexity in plausibility assessment task performance indicates that mastery in this domain necessitates a detailed understanding of the interplay among model architecture, size, and training strategies.

**While variations in performance across prompts and metrics are observed, they typically correlate strongly.** The most significant gap occur between OLMo-1B's performance on the text completion prompt (22.11 ACC.) and the question answering prompt (42.15 ACC.), highlighting some variation in model proficiency across different types of tasks. Despite these variations, the assessment appears to hinge on a consistent underlying mechanism across prompt types. This is further evidenced by the Pearson correlation coefficients be-

| MODEL | PROBELM PLAUSIBILITY | COPA ACC. | ARC ACC.-NORM | HELLASWAG ACC.-NORM | TRUTHFULQA ACC. | WINOGRANDE ACC. |
|---|---|---|---|---|---|---|
| PYTHIA-2.8B | 58.47 (1) | 79.0 (4) | 33.02 (4) | 59.30 (5) | 35.88 (7) | 59.12 (5) |
| OLMO-7B | 52.51 (2) | 85.0 (1) | 40.36 (2) | 75.65 (2) | 35.85 (8) | 66.38 (2) |
| GPT-2-1.5B | 45.51 (3) | 76.0 (5) | 28.50 (6) | 50.89 (6) | 38.53 (6) | 58.33 (6) |
| OLMO-1B | 45.07 (4) | 82.0 (3) | 31.06 (5) | 62.92 (3) | 32.94 (10) | 59.98 (4) |
| PYTHIA-160M | 42.45 (5) | 64.0 (7) | 23.21 (8) | 28.48 (9) | 44.43 (2) | 49.88 (10) |
| LLAMA-7B | 40.74 (6) | 85.0 (1) | 44.62 (1) | 76.21 (2) | 34.08 (9) | 70.01 (1) |
| BLOOM-7B | 40.74 (7) | 73.0 (6) | 33.45 (3) | 62.28 (4) | 38.89 (5) | 64.64 (3) |
| GPT-2-124M | 39.02 (8) | 62.0 (8) | 22.70 (9) | 31.14 (8) | 40.69 (4) | 51.62 (7) |
| PYTHIA-14M | 35.90 (9) | 53.0 (10) | 21.25 (10) | 26.12 (10) | 50.37 (1) | 50.43 (9) |
| BLOOM-560M | 30.63 (10) | 61.0 (9) | 23.81 (7) | 36.92 (7) | 42.43 (3) | 51.38 (8) |

Table 3: Language Models Performance on PROBELM versus other reasoning datasets. The **ranking** of the models is shown in brackets. ARC shows results on the challenge set. Same as before, *plausibility* shows the average score of all prompt types and metrics (ACC., MRR, and NDCG). Models are ordered based on the plausibility rank on PROBELM.

tween the performances on various prompts, which were notably high: 0.8281 for statement versus text completion, 0.8672 for statement versus question answering, and 0.8976 for text completion versus question answering. Such strong correlations underscore that, despite the prompt-specific performance disparities, the models are evaluated based on a core aspect of their capability of ranking plausibility. Similarly, the Pearson correlation score between MRR and NDCG is 0.9982, indicating consistent plausibility measurements across metrics.

## 5.2 Comparison with Other Benchmarks

To comprehensively understand the different coverage between PROBELM and other LLM benchmarks, we follow `lm-evaluation-harness`[5] to evaluate zero-shot performance on COPA, TruthfulQA (mc2, i.e., multiple options can be correct), ARC (challenge set) (Clark et al., 2018), HellaSwag (Zellers et al., 2019), and Winogrande (Sakaguchi et al., 2021). Default metrics (accuracy or normalised accuracy) are applied to each dataset and the results are included in Table 3.

**PROBELM shows distinct model performance compared to other reasoning benchmarks**. Models that perform well on tasks emphasising factual accuracy do not always maintain their performance levels on plausibility assessments. In some cases, the performance rankings are inversely related. A notable example of this phenomenon is observed in the performance disparity between PROBELM and TruthfulQA. Specifically, Pythia-14M, which exhibits the best performance in TruthfulQA, positions near the bottom (9th place) of the PROBELM ranking. This variance underscores the distinct nature of plausibility as an evaluation criterion, demanding separate capabilities beyond those required for tasks focused on factual accuracy. The development of PROBELM thus contributes to the broader framework for evaluating language models by offering a different angle to assess their capability to understand and generate plausible content.

## 5.3 Temporal Effect of PROBELM

We further evaluate models against datasets from 2017, 2020, 2021, and 2023 timestamps to investigate how the recency of training data affects the plausibility performance. The outcomes of this analysis are presented in Figure 2 and detailed in Appendix C. Remarkably, the relative rankings of models prove relatively consistent across these varied timeframes, affirming the robustness of our evaluation approach. We observe that, from the 2020 to 2023 timestamps, models trained closer to or after the evaluation dates typically exhibit enhanced performance, underscoring the benefits of leveraging recent data.

---

[5]https://github.com/EleutherAI/lm-evaluation-harness

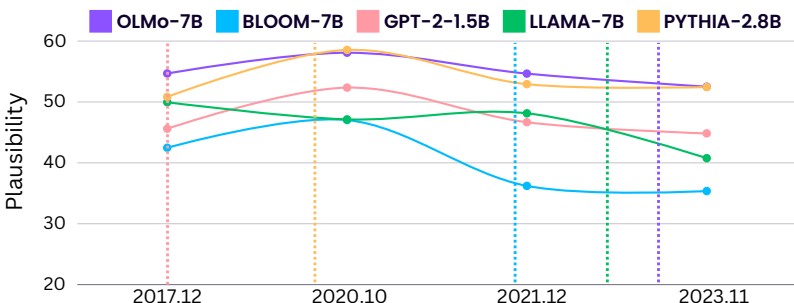

Figure 2: PLAUSIBILITY scores of models on the timeframe specific datasets. We illustrate one model per family. Vertical dotted lines represent the training timeframe of each model.

An anomaly is detected in the 2017 dataset, where all models show a decrease in performance, suggesting it is an outlier. An analysis reveals that the "occupation" relation disproportionally influences the 2017 dataset, constituting 14.3% of all scenarios compared to less than 0.5% in the other datasets. This discrepancy indicates a significant impact of the "occupation" relation on model performance. Excluding the "occupation" relation from the evaluation on the 2017 dataset results in either similar or higher scores across models, shown in Appendix C. For instance, evaluating the Pythia-160M model without scenarios involving the "occupation" relation leads to a notable increase in plausibility scores from 39.69 to 42.73. This outcome indicates the models' challenges with certain relations.

## 6 Conclusion

In this work, we introduce PROBELM, a benchmark designed to evaluate language models' ability to discern plausible scenarios, filling a critical gap left by existing benchmarks focused on factual accuracy or plausibility without world knowledge integration. Our comprehensive evaluation of 10 models of varying sizes and architectures against this benchmark reveals key insights including factual accuracy does not directly correlate with plausibility discernment; larger models do not uniformly outperform smaller ones across different families; the recency of training data, while impactful, is not the sole determinant of plausibility performance; and models tend to struggle with plausibility assessment as the gap between their training data and the evaluation set widens. These findings highlight the complexity of plausibility inference and the need for advanced modelling techniques that can better address the challenges of real-world knowledge application. Our dataset and code are publicly available, to facilitate the community for further exploration in enhancing language models' plausibility inference capabilities.

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

## A  Models Used in Our Experiments

We briefly introduce below the models and the timestamp of the corresponding Wikidata used for our experiments.

**GPT-2 124M and 1.5B**  (Radford et al., 2019) As early iterations in the GPT series, these models offer a perspective on plausibility inference with training data frozen in December 2017. This provides a basis for evaluating how well models can reason about plausibility without the benefit of recent information. The closest corresponding timestamp we found is from December 2017 to January 2018.

**BLOOM 560M and 7B**  (Scao et al., 2023) Representing the BLOOM series' commitment to openness and scalability, these models, with data up to December 2021, allow us to explore how differences in model size impact their capacity to discern plausible scenarios without having been exposed to subsequent updates. The closest corresponding timestamp we found is from December 2021 to January 2022.

**LLaMA 7B**  (Touvron et al., 2023) LLaMA is recognised for its robust performance across a broad range of NLP tasks. Its training data, current as of August 2022, positions it uniquely for evaluation with the PROBELM benchmark, specifically designed to assess the model's ability to infer plausibility based on its training up to that point. The closest corresponding timestamp we found is from December 2023 to January 2023.

**Pythia 14M, 160M, and 2.8B**  (Biderman et al., 2023) The Pythia models extend our exploration into the realms of modern architecture and model scalability, spanning the spectrum from an ultra-compact 14M to a robust 2.8B parameter setup. This inclusion allows us to assess plausibility inference capabilities across a broad range of model sizes, particularly emphasising the performance of smaller models like the Pythia 14M. The evaluation of Pythia models leverages their training data up to March 2020. The closest corresponding timestamp we found is from October 2020 to November 2020.

**OLMo 1B and 7B**  (Groeneveld et al., 2024) The newest models in our evaluation, OLMo's training data extends to March 2023. Their assessment against the PROBELM benchmark tests their ability to gauge plausibility based on a comprehensive and current dataset, without incorporating the very latest updates. The closest corresponding timestamp we found is from December 2023 to January 2023.

## B  Algorithm for Generating Less Plausible Scenarios

To systematically generate less plausible scenarios for the PROBELM benchmark, we use the algorithm shown in Algorithm 1, which leverages the statistical distributions of entity co-occurrences within Wikidata. This method ensures that the generated scenarios, while being less plausible, still maintain logical coherence and relevance to the given context.

---

**Algorithm 1** Generating Less Plausible Scenarios

---

**Require:** A triple from Wikidata $(s, r, o)$
**Ensure:** Ranked list of less plausible triples $L$
  $r$ remains constant for all operations
  $P \leftarrow \{p \mid (s, \text{subclass\_of}, p) \in \text{Wikidata}\}$
  $S \leftarrow$ empty list to hold siblings of $s$
  **for all** entities $e$ sharing parent $P$ **do**
    **if** $e \neq s$ **then**
      Append $e$ to $S$
    **end if**
  **end for**
  $O_{all} \leftarrow$ empty list to hold objects for all siblings
  **for all** $sib \in S$ **do**
    $O_{sib} \leftarrow$ all objects for triples $(sib, r, *)$
    Append $O_{sib}$ to $O_{all}$
  **end for**
  Initialize an empty map $Freq$
  **for all** $obj \in O_{all}$ **do**
    **if** $obj$ in $Freq$ **then**
      $Freq[obj] \leftarrow Freq[obj] + 1$
    **else**
      $Freq[obj] \leftarrow 1$
    **end if**
  **end for**
  $L \leftarrow \text{sort}(Freq, \text{key} = Freq.\text{get}, \text{reverse})$
  Repeat process with $o$ fixed to find subjects
  similar to $s$ and generate another ranked list $L$

---

## C   Additional Results on Time-Specific Datasets

We show the plausibility performance for models evaluated on PROBELM with different timestamps in Table 4 (2017), Table 5 (2020), Table 6 (2021), and Table 7 (2023).

| MODEL | AVG | Statement | | | Text Completion | | | Question Answering | | |
|---|---|---|---|---|---|---|---|---|---|---|
| | PLAUSIBILITY | ACC. | MRR | NDCG | ACC. | MRR | NDCG | ACC. | MRR | NDCG |
| OLMO-1B | 46.64 (49.91) | 30.52 | 48.90 | 47.57 | 26.75 | 49.72 | 48.78 | 44.28 | 62.05 | 61.21 |
| OLMO-7B | **54.68** (57.72) | 33.41 | 52.70 | 51.76 | **46.73** | **64.68** | **64.37** | **47.84** | **65.86** | **64.76** |
| BLOOM-560M | 37.95 (37.94) | 23.20 | 46.05 | 45.35 | 20.76 | 43.82 | 43.40 | 24.31 | 47.67 | 47.01 |
| BLOOM-7B | 42.48 (43.02) | 29.86 | 50.01 | 49.39 | 28.86 | 52.22 | 51.78 | 23.31 | 48.72 | 48.18 |
| GPT-2-124M | 39.02 (42.83) | 29.08 | 49.08 | 47.96 | 18.09 | 40.84 | 39.83 | 26.97 | 50.12 | 49.19 |
| GPT-2-1.5B | 45.51 (50.16) | 29.19 | 49.81 | 48.83 | 25.52 | 48.92 | 48.11 | 41.95 | 58.92 | 58.34 |
| LLAMA-7B | 49.96 (48.38) | 30.52 | 51.70 | 51.04 | 40.95 | 61.01 | 60.37 | 37.63 | 58.57 | 49.96 |
| PYTHIA-14M | 34.03 (37.77) | 20.98 | 41.50 | 40.30 | 16.09 | 37.52 | 36.31 | 22.97 | 45.94 | 44.65 |
| PYTHIA-160M | 39.69 (42.73) | 30.30 | 50.27 | 49.51 | 15.54 | 37.08 | 36.15 | 30.97 | 53.75 | 53.62 |
| PYTHIA-2.8B | 50.85 (54.73) | **34.30** | **53.08** | **51.98** | 36.51 | 58.44 | 58.37 | 42.51 | 61.69 | 60.80 |

Table 4: Evaluation Results of Language Models on PROBELM Using the Same Evaluation Dataset of December 2017 to January 2018. Plausibility Scores in Parenthesis Are Obtained Without Evaluating the "Occupation" Relation.

## D   More details on PROBELM

An example of the prompts used to evaluate the models is shown in Table 8.

| MODEL | AVG | Statement | | | Text Completion | | | Question Answering | | |
|-------|-----|-----------|---|---|-----------------|---|---|--------------------|---|---|
| | PLAUSIBILITY | ACC. | MRR | NDCG | ACC. | MRR | NDCG | ACC. | MRR | NDCG |
| OLMO-1B | 53.59 | 35.74 | 54.94 | 55.41 | 32.93 | 56.40 | 57.20 | 52.61 | 68.19 | **69.11** |
| OLMO-7B | _58.09_ | _39.36_ | 57.53 | 57.25 | **49.60** | **67.22** | **66.31** | 50.40 | 67.43 | 67.07 |
| BLOOM-560M | 40.93 | 31.12 | 53.08 | 53.17 | 17.87 | 44.27 | 44.40 | 23.89 | 50.21 | 50.23 |
| BLOOM-7B | 47.03 | 39.16 | _58.70_ | _59.34_ | 28.71 | 54.87 | 55.23 | 23.69 | 51.60 | 51.81 |
| GPT-2-124M | 42.96 | 32.73 | 52.78 | 52.99 | 18.47 | 43.58 | 43.72 | 30.72 | 55.54 | 56.11 |
| GPT-2-1.5B | 52.36 | 37.35 | 57.67 | 57.74 | 26.70 | 52.22 | 52.36 | _52.81_ | _68.37_ | 68.61 |
| LLAMA-7B | 47.14 | 25.30 | 48.46 | 48.69 | 35.74 | 57.89 | 58.25 | 34.33 | 57.61 | 57.95 |
| PYTHIA-14M | 35.89 | 20.68 | 43.15 | 43.11 | 17.67 | 39.48 | 39.37 | 23.90 | 47.90 | 47.85 |
| PYTHIA-160M | 42.45 | 34.54 | 54.39 | 55.03 | 16.87 | 39.59 | 40.13 | 30.52 | 55.16 | 55.84 |
| PYTHIA-2.8B | **58.55** | **43.78** | **61.17** | **61.14** | _43.57_ | _62.77_ | _62.33_ | **53.41** | **69.13** | _68.89_ |

Table 5: Evaluation Results of Language Models on PROBELM Using the Same Evaluation Dataset of October to November 2020.

| MODEL | AVG | Statement | | | Text Completion | | | Question Answering | | |
|-------|-----|-----------|---|---|-----------------|---|---|--------------------|---|---|
| | PLAUSIBILITY | ACC. | MRR | NDCG | ACC. | MRR | NDCG | ACC. | MRR | NDCG |
| OLMO-1B | 50.22 | _38.52_ | 56.12 | 55.17 | 28.73 | 51.99 | 51.24 | 45.92 | 63.21 | 62.12 |
| OLMO-7B | **54.66** | 38.02 | _56.87_ | _55.83_ | **44.29** | **62.85** | **61.73** | **45.92** | **63.81** | **62.63** |
| BLOOM-560M | 30.63 | 20.58 | 43.37 | 42.56 | 8.78 | 34.81 | 34.54 | 12.54 | 39.44 | 39.02 |
| BLOOM-7B | 36.22 | 27.35 | 49.90 | 49.50 | 17.44 | 44.21 | 43.39 | 12.80 | 41.11 | 40.29 |
| GPT-2-124M | 38.92 | 32.25 | 51.89 | 51.22 | 14.81 | 39.64 | 39.38 | 24.22 | 48.82 | 48.25 |
| GPT-2-1.5B | 46.68 | 35.51 | 56.02 | 54.81 | 19.70 | 45.68 | 45.06 | 43.66 | 60.86 | 59.86 |
| LLAMA-7B | 48.14 | 28.48 | 51.28 | 50.22 | 35.88 | 57.62 | 56.14 | 38.14 | 58.60 | 56.86 |
| PYTHIA-14M | 35.16 | 20.08 | 42.90 | 43.15 | 15.55 | 38.66 | 39.38 | 22.08 | 47.06 | 47.69 |
| PYTHIA-160M | 36.66 | 31.37 | 53.11 | 52.37 | 11.92 | 35.31 | 34.82 | 24.21 | 48.87 | 48.71 |
| PYTHIA-2.8B | _52.93_ | **41.53** | **58.64** | **57.78** | 37.64 | 57.46 | 56.60 | 44.54 | 61.86 | 60.93 |

Table 6: Evaluation Results of Language Models on PROBELM Using the Same Evaluation Dataset of December 2021 to January 2022.

We show the relations included in PROBELM and different types of prompts in Table 9, Table 10, and Table 11.

| MODEL | AVG PLAUSIBILITY | Statement | | | Text Completion | | | Question Answering | | |
|---|---|---|---|---|---|---|---|---|---|---|
| | | ACC. | MRR | NDCG | ACC. | MRR | NDCG | ACC. | MRR | NDCG |
| OLMO-1B | 45.07 | 29.96 | 49.92 | 50.13 | 22.11 | 46.47 | 46.95 | 42.15 | 59.09 | 58.89 |
| OLMO-7B | **52.51** | 31.20 | 51.62 | 51.57 | **44.62** | **61.87** | **61.43** | 45.45 | 62.60 | 62.23 |
| BLOOM-560M | 32.07 | 22.93 | 46.51 | 46.49 | 7.85 | 36.55 | 37.04 | 10.33 | 39.90 | 40.03 |
| BLOOM-7B | 35.37 | 30.79 | 51.56 | 51.34 | 13.42 | 42.81 | 42.44 | 8.47 | 38.91 | 38.58 |
| GPT-2-124M | 35.06 | 23.97 | 46.22 | 46.72 | 10.95 | 36.35 | 36.79 | 20.45 | 46.87 | 47.18 |
| GPT-2-1.5B | 44.84 | 29.75 | 51.70 | 50.92 | 20.25 | 46.21 | 46.05 | 41.12 | 59.18 | 58.59 |
| LLAMA-7B | 40.77 | 19.42 | 43.26 | 43.60 | 29.96 | 52.08 | 51.81 | 27.69 | 49.56 | 49.51 |
| PYTHIA-14M | 28.89 | 12.40 | 37.21 | 37.74 | 9.30 | 32.76 | 33.77 | 14.67 | 41.02 | 41.14 |
| PYTHIA-160M | 36.34 | 26.86 | 49.87 | 50.18 | 10.95 | 34.04 | 34.99 | 22.52 | 48.60 | 49.05 |
| PYTHIA-2.8B | 52.46 | **36.16** | **55.55** | **55.34** | 36.98 | 58.42 | 58.26 | **46.07** | **63.45** | **62.86** |

Table 7: Evaluation Results of Language Models on PROBELM Using the Same Evaluation Dataset of November to December 2023.

| Prompt Type | Examples |
|---|---|
| STATEMENT | "Marcelo is Spanish.", "Marcelo is Irish.", "Marcelo is Chinese." |
| TEXT COMPLETION | "Fill in the blank: Marcelo has citizenship of ____. Answer: Spanish." |
| QUESTION ANSWERING | "Question: What citizenship(s) does Marcelo have? Answer: Spanish." |

Table 8: Illustrative Examples of Prompt Types Used in PROBELM.

| PID | RELATION | TEMPLATE FOR STATEMENT |
|---|---|---|
| P5096 | member of the crew of | \<subject\>is a member of the crew of \<object\>. |
| P122 | basic form of government | The basic form of government of \<subject\>is \<object\>. |
| P3448 | stepparent | \<subject\>is the stepparent of \<object\>. |
| P1479 | has contributing factor | \<object\>is a contributing factor to \<subject\>. |
| P61 | discoverer or inventor | \<subject\>is the discoverer or inventor of \<object\>. |
| P3320 | board member | \<subject\>serves as a board member of \<object\>. |
| P7779 | member of military unit | \<subject\>is a member of the military unit \<object\>. |
| P98 | editor | \<subject\>is the editor of \<object\>. |
| P1411 | nominated for | \<subject\>was nominated for \<object\>. |
| P371 | presenter | \<subject\>is the presenter of \<object\>. |
| P1365 | replaces | \<subject\>replaces \<object\>. |
| P488 | chairperson | \<subject\>serves as the chairperson of \<object\>. |
| P27 | country of citizenship | \<subject\>'s country of citizenship is \<object\>. |
| P20 | place of death | \<subject\>'s place of death is \<object\>. |
| P1344 | participant in | \<subject\>was a participant in \<object\>. |
| P1366 | replaced by | \<subject\>was replaced by \<object\>. |
| P1412 | languages spoken written or signed | \<subject\>speaks the following languages: \<object\>. |
| P276 | location | The location of \<subject\>is \<object\>. |
| P407 | language of work or name | The language of \<subject\>'s work or name is \<object\>. |
| P39 | position held | \<subject\>holds the position of \<object\>. |
| P1532 | country for sport | \<subject\>represents \<object\>in sports competitions. |
| P451 | unmarried partner | \<subject\>is the unmarried partner of \<object\>. |
| P54 | member of sports team | \<subject\>is a member of the sports team \<object\>. |
| P800 | notable work | \<subject\>'s notable work includes \<object\>. |
| P551 | residence | \<subject\>'s residence is in \<object\>. |
| P131 | located in the administrative territorial entity | \<subject\>is located in the administrative territorial entity \<object\>. |
| P106 | occupation | \<subject\>'s occupation is \<object\>. |
| P69 | educated at | \<subject\>was educated at \<object\>. |
| P509 | cause of death | \<subject\>'s cause of death was \<object\>. |
| P102 | member of political party | \<subject\>is a member of the political party \<object\>. |
| P19 | place of birth | \<subject\>'s place of birth is \<object\>. |
| P115 | home venue | \<subject\>'s home venue is \<object\>. |
| P1001 | applies to jurisdiction | \<object\>applies to the jurisdiction of \<subject\>. |
| P840 | narrative location | \<subject\>is set in the narrative location of \<object\>. |
| P108 | employer | \<subject\>is employed by \<object\>. |
| P57 | director | \<subject\>is the director of \<object\>. |
| P2416 | sports discipline competed in | \<subject\>competes in the sports discipline of \<object\>. |
| P400 | platform | \<subject\>is available on the platform \<object\>. |
| P1433 | published in | \<subject\>was published in \<object\>. |
| P1056 | product or material produced | \<subject\>produces \<object\>as a product or material. |
| P9071 | character type | \<subject\>is characterized as a \<object\>type. |
| P4100 | parliamentary group | \<subject\>is a member of the parliamentary group \<object\>. |
| P937 | work location | \<subject\>'s work location is \<object\>. |
| P1066 | student of | \<subject\>is a student of \<object\>. |
| P1535 | used by | \<object\>is used by \<subject\>. |
| P6 | head of government | \<subject\>is the head of government of \<object\>. |
| P2283 | use | \<subject\>is used for \<object\>. |
| P812 | academic major | \<subject\>'s academic major is \<object\>. |
| P1416 | affiliation | \<subject\>is affiliated with \<object\>. |
| P2522 | victory | \<subject\>achieved a victory in \<object\>. |
| P607 | conflict | \<subject\>was involved in the conflict \<object\>. |
| P749 | parent organization | \<subject\>is a part of the parent organization \<object\>. |
| P2283 | uses | \<subject\>uses \<object\>. |
| P802 | student | \<subject\>is a student at \<object\>. |
| P119 | place of burial | \<subject\>'s place of burial is \<object\>. |
| P2842 | place of marriage | \<subject\>was married at \<object\>. |
| P286 | head coach | \<subject\>is the head coach of \<object\>. |
| P2541 | operating area | \<subject\>'s operating area is \<object\>. |
| P1441 | present in work | \<subject\>is present in the work \<object\>. |
| P2650 | interested in | \<subject\>is interested in \<object\>. |
| P1027 | conferred by | \<object\>is conferred by \<subject\>. |
| P3300 | musical conductor | \<subject\>is the musical conductor of \<object\>. |
| P2715 | elected in | \<subject\>was elected in \<object\>. |
| P2937 | parliamentary term | \<subject\>served during the parliamentary term \<object\>. |
| P1399 | convicted of | \<subject\>was convicted of \<object\>. |
| P1686 | for work | \<subject\>is used for the work \<object\>. |
| P1196 | manner of death | \<subject\>'s manner of death was \<object\>. |
| P2632 | place of detention | \<subject\>was detained at \<object\>. |
| P991 | successful candidate | \<subject\>was the successful candidate in \<object\>. |
| P2443 | stage reached | \<subject\>reached the stage \<object\>. |
| P6872 | has written for | \<subject\>has written for \<object\>. |

Table 9: Relations and templates for *statement* prompts used in PROBELM.

| PID | RELATION | TEMPLATE FOR **TEXT COMPLETION** |
|---|---|---|
| P5096 | member of the crew of | Fill in the blank: <subject>is a member of the crew of ____. Answer: <object> |
| P122 | basic form of government | Fill in the blank: The basic form of government of <subject>is ____. Answer: <object> |
| P3448 | stepparent | Fill in the blank: <subject>is the stepparent of ____. Answer: <object> |
| P1479 | has contributing factor | Fill in the blank: <object>is a contributing factor to ____. Answer: <subject> |
| P61 | discoverer or inventor | Fill in the blank: ____ is the discoverer or inventor of <object>. Answer: <subject> |
| P3320 | board member | Fill in the blank: <subject>serves as a board member of ____. Answer: <object> |
| P7779 | member of military unit | Fill in the blank: <subject>is a member of the military unit ____. Answer: <object> |
| P98 | editor | Fill in the blank: ____ is the editor of <object>. Answer: <subject> |
| P1411 | nominated for | Fill in the blank: ____ was nominated for <object>. Answer: <subject> |
| P371 | presenter | Fill in the blank: ____ is the presenter of <object>. Answer: <subject> |
| P1365 | replaces | Fill in the blank: <subject>replaces ____. Answer: <object> |
| P488 | chairperson | Fill in the blank: ____ serves as the chairperson of <object>. Answer: <subject> |
| P27 | country of citizenship | Fill in the blank: <subject>'s country of citizenship is ____. Answer: <object> |
| P20 | place of death | Fill in the blank: <subject>'s place of death is ____. Answer: <object> |
| P1344 | participant in | Fill in the blank: ____ was a participant in <object>. Answer: <subject> |
| P1366 | replaced by | Fill in the blank: <subject>was replaced by ____. Answer: <object> |
| P1412 | languages spoken written or signed | Fill in the blank: <subject>speaks the following languages: ____. Answer: <object> |
| P276 | location | Fill in the blank: The location of <subject>is ____. Answer: <object> |
| P407 | language of work or name | Fill in the blank: The language of <subject>'s work or name is ____. Answer: <object> |
| P39 | position held | Fill in the blank: <subject>holds the position of ____. Answer: <object> |
| P1532 | country for sport | Fill in the blank: ____ represents <object>in sports competitions. Answer: <subject> |
| P451 | unmarried partner | Fill in the blank: <subject>is the unmarried partner of ____. Answer: <object> |
| P54 | member of sports team | Fill in the blank: <subject>is a member of the sports team ____. Answer: <object> |
| P800 | notable work | Fill in the blank: <subject>'s notable work includes ____. Answer: <object> |
| P551 | residence | Fill in the blank: <subject>'s residence is in ____. Answer: <object> |
| P131 | located in the administrative territorial entity | Fill in the blank: <subject>is located in the administrative territorial entity ____. Answer: <object> |
| P106 | occupation | Fill in the blank: <subject>'s occupation is ____. Answer: <object> |
| P69 | educated at | Fill in the blank: <subject>was educated at ____. Answer: <object> |
| P509 | cause of death | Fill in the blank: <subject>'s cause of death was ____. Answer: <object> |
| P102 | member of political party | Fill in the blank: <subject>is a member of the political party ____. Answer: <object> |
| P19 | place of birth | Fill in the blank: <subject>'s place of birth is ____. Answer: <object> |
| P115 | home venue | Fill in the blank: <subject>'s home venue is ____. Answer: <object> |
| P1001 | applies to jurisdiction | Fill in the blank: <object>applies to the jurisdiction of ____. Answer: <subject> |
| P840 | narrative location | Fill in the blank: <subject>is set in the narrative location of ____. Answer: <object> |
| P108 | employer | Fill in the blank: <subject>is employed by ____. Answer: <object> |
| P57 | director | Fill in the blank: ____ is the director of <object>. Answer: <subject> |
| P2416 | sports discipline competed in | Fill in the blank: <subject>competes in the sports discipline of ____. Answer: <object> |
| P400 | platform | Fill in the blank: <subject>is available on the platform ____. Answer: <object> |
| P1433 | published in | Fill in the blank: <subject>was published in ____. Answer: <object> |
| P1056 | product or material produced | Fill in the blank: <subject>produces ____ as a product or material. Answer: <object> |
| P9071 | character type | Fill in the blank: <subject>is characterized as a ____ type. Answer: <object> |
| P4100 | parliamentary group | Fill in the blank: <subject>is a member of the parliamentary group ____. Answer: <object> |
| P937 | work location | Fill in the blank: <subject>'s work location is ____. Answer: <object> |
| P1066 | student of | Fill in the blank: <subject>is a student of ____. Answer: <object> |
| P1535 | used by | Fill in the blank: <object>is used by ____. Answer: <subject> |
| P6 | head of government | Fill in the blank: ____ is the head of government of <object>. Answer: <subject> |
| P2283 | use | Fill in the blank: <subject>is used for ____. Answer: <object> |
| P812 | academic major | Fill in the blank: <subject>'s academic major is ____. Answer: <object> |
| P1416 | affiliation | Fill in the blank: <subject>is affiliated with ____. Answer: <object> |
| P2522 | victory | Fill in the blank: <subject>achieved a victory in ____. Answer: <object> |
| P607 | conflict | Fill in the blank: <subject>was involved in the conflict ____. Answer: <object> |
| P749 | parent organization | Fill in the blank: <subject>is a part of the parent organization ____. Answer: <object> |
| P2283 | uses | Fill in the blank: <subject>uses ____. Answer: <object> |
| P802 | student | Fill in the blank: <subject>is a student at ____. Answer: <object> |
| P119 | place of burial | Fill in the blank: <subject>'s place of burial is ____. Answer: <object> |
| P2842 | place of marriage | Fill in the blank: <subject>was married at ____. Answer: <object> |
| P286 | head coach | Fill in the blank: ____ is the head coach of <object>. Answer: <subject> |
| P2541 | operating area | Fill in the blank: <subject>'s operating area is ____. Answer: <object> |
| P1441 | present in work | Fill in the blank: <subject>is present in the work ____. Answer: <object> |
| P2650 | interested in | Fill in the blank: <subject>is interested in ____. Answer: <object> |
| P1027 | conferred by | Fill in the blank: <object>is conferred by ____. Answer: <subject> |
| P3300 | musical conductor | Fill in the blank: ____ is the musical conductor of <object>. Answer: <subject> |
| P2715 | elected in | Fill in the blank: <subject>was elected in <object>. Answer: <object> |
| P2937 | parliamentary term | Fill in the blank: <subject>served during the parliamentary term ____. Answer: <object> |
| P1399 | convicted of | Fill in the blank: <subject>was convicted of ____. Answer: <object> |
| P1686 | for work | Fill in the blank: <subject>is used for the work ____. Answer: <object> |
| P1196 | manner of death | Fill in the blank: <subject>'s manner of death was ____. Answer: <object> |
| P2632 | place of detention | Fill in the blank: <subject>was detained at ____. Answer: <object> |
| P991 | successful candidate | Fill in the blank: <subject>was the successful candidate in ____. Answer: <object> |
| P2443 | stage reached | Fill in the blank: <subject>reached the stage ____. Answer: <object> |
| P6872 | has written for | Fill in the blank: <subject>has written for ____. Answer: <object> |

Table 10: Relations and templates for *text completion* prompts used in PROBELM.

| PID | RELATION | TEMPLATE FOR **QUESTION ANSWERING** |
|---|---|---|
| P5096 | member of the crew of | Question: <subject>is a member of the crew of what? Answer: <object>. |
| P122 | basic form of government | Question: What is the basic form of government of <subject>? Answer: <object>. |
| P3448 | stepparent | Question: <subject>is the stepparent of who? Answer: <object>. |
| P1479 | has contributing factor | Question: <object>is a contributing factor to What? Answer: <subject>. |
| P61 | discoverer or inventor | Question: Who is the discoverer or inventor of <object>? Answer: <subject>. |
| P3320 | board member | Question: <subject>serves as a board member of what? Answer: <object>. |
| P7779 | member of military unit | Question: <subject>is a member of the military unit of what? Answer: <object>. |
| P98 | editor | Question: Who is is the editor of <object>? Answer: <subject>. |
| P1411 | nominated for | Question: Who was nominated for <object>? Answer: <subject>. |
| P371 | presenter | Question: Who is the presenter of <object>? Answer: <subject>. |
| P1365 | replaces | Question: Whom did <subject>replace? Answer: <object>. |
| P488 | chairperson | Question: Who serves as the chairperson of <object>? Answer: <subject>. |
| P27 | country of citizenship | Question: What is <subject>'s country of citizenship? Answer: <object>. |
| P20 | place of death | Question: Where is <subject>'s place of death? Answer: <object>. |
| P1344 | participant in | Question: Who was a participant in <object>? Answer: <subject>. |
| P1366 | replaced by | Question: What was <subject>replaced by? Answer: <object>. |
| P1412 | languages spoken written or signed | Question: What language does <subject>speaks? Answer: <object>. |
| P276 | location | Question: Where is the location of <subject>? Answer: <object>. |
| P407 | language of work or name | Question: What is the language of <subject>'s work or name? Answer: <object>. |
| P39 | position held | Question: What position does <subject>hold? Answer: <object>. |
| P1532 | country for sport | Question: Who represents <object>in sports competitions? Answer: <subject>. |
| P451 | unmarried partner | Question: Who is the unmarried partner of <subject>? Answer: <object>. |
| P54 | member of sports team | Question: <subject>is a member of which sports team? Answer: <object>. |
| P800 | notable work | Question: What is a notable work of <subject>? Answer: <object>. |
| P551 | residence | Question: Where is the residence of <subject>? Answer: <object>. |
| P131 | located in the administrative territorial entity | Question: In which administrative territorial entity is <subject>located? Answer: <object>. |
| P106 | occupation | Question: What is the occupation of <subject>? Answer: <object>. |
| P69 | educated at | Question: Where was <subject>educated? Answer: <object>. |
| P509 | cause of death | Question: What was the cause of death of <subject>? Answer: <object>. |
| P102 | member of political party | Question: Which political party is <subject>a member of? Answer: <object>. |
| P19 | place of birth | Question: Where is the place of birth of <subject>? Answer: <object>. |
| P115 | home venue | Question: What is the home venue of <subject>? Answer: <object>. |
| P1001 | applies to jurisdiction | Question: To which jurisdiction does <object>apply? Answer: <subject>. |
| P840 | narrative location | Question: What is the narrative location of <subject>? Answer: <object>. |
| P108 | employer | Question: Who is the employer of <subject>? Answer: <object>. |
| P57 | director | Question: Who is the director of <object>? Answer: <subject>. |
| P2416 | sports discipline competed in | Question: In which sports discipline does <subject>compete? Answer: <object>. |
| P400 | platform | Question: On which platform is <subject>available? Answer: <object>. |
| P1433 | published in | Question: Where was <subject>published? Answer: <object>. |
| P1056 | product or material produced | Question: What product or material does <subject>produce? Answer: <object>. |
| P9071 | character type | Question: What type of character is <subject>characterized as? Answer: <object>. |
| P4100 | parliamentary group | Question: Which parliamentary group is <subject>a member of? Answer: <object>. |
| P937 | work location | Question: What is the work location of <subject>? Answer: <object>. |
| P1066 | student of | Question: Who is <subject>a student of? Answer: <object>. |
| P1535 | used by | Question: Who uses <object>? Answer: <subject>. |
| P6 | head of government | Question: Who is the head of government of <object>? Answer: <subject>. |
| P2283 | use | Question: What is <subject>used for? Answer: <object>. |
| P812 | academic major | Question: What is <subject>'s academic major?Answer: <object>. |
| P1416 | affiliation | Question: What is <subject>affiliated with? Answer: <object>. |
| P2522 | victory | Question: In which <object>did <subject>achieve a victory? Answer: <object>. |
| P607 | conflict | Question: In which conflict was <subject>involved? Answer: <object>. |
| P749 | parent organization | Question: What parent organization is <subject>a part of? Answer: <object>. |
| P2283 | uses | Question: What does <subject>use? Answer: <object>. |
| P802 | student | Question: At which institution is <subject>a student? Answer: <object>. |
| P119 | place of burial | Question: Where is <subject>buried? Answer: <object>. |
| P2842 | place of marriage | Question: Where was <subject>married? Answer: <object>. |
| P286 | head coach | Question: Who is the head coach of <object>? Answer: <subject>. |
| P2541 | operating area | Question: What is the operating area of <subject>? Answer: <object>. |
| P1441 | present in work | Question: In which work is <subject>present? Answer: <object>. |
| P2650 | interested in | Question: What is <subject>interested in? Answer: <object>. |
| P1027 | conferred by | Question: Who conferred <object>? Answer: <subject>. |
| P3300 | musical conductor | Question: Who is the musical conductor of <object>? Answer: <subject>. |
| P2715 | elected in | Question: In what was <subject>elected? Answer: <object>. |
| P2937 | parliamentary term | Question: During which parliamentary term did <subject>serve? Answer: <object>. |
| P1399 | convicted of | Question: What was <subject>convicted of? Answer: <object>. |
| P1686 | for work | Question: What is <subject>used for in terms of work? Answer: <object>. |
| P1196 | manner of death | Question: What was the manner of death of <subject>? Answer: <object>. |
| P2632 | place of detention | Question: Where was <subject>detained? Answer: <object>. |
| P991 | successful candidate | Question: In which election or selection was <subject>the successful candidate? Answer: <object>. |
| P2443 | stage reached | Question: What stage did <subject>reach? Answer: <object>. |
| P6872 | has written for | Question: For whom or what has <subject>written? Answer: <object>. |

Table 11: Relations and templates for *question answering* prompts used in PRoBELM.

