# OpenReview forum: "PRobELM: Plausibility Ranking Evaluation for Language Models"
_colmweb.org/COLM/2024/Conference — COLM_

### Official Review · Reviewer_9vFJ · 2024-05-06

**Rating:** 8
**Confidence:** 3
**Ethics Flag:** 1

**Summary:**

This paper introduces a new benchmark called PRobELM, which aims at testing the LLMs capacity of recognizing plausibility of events and scenarios. PRobELM is created from facts extracted from WikiData, using data points that were added at a later timestamp than the training data of the models under evaluation, and then a set of alternative, less plausible facts are generated for each of them and LLMs are asked to rank them from the most to the least plausible.

The authors evaluate the new benchmark with 10 different LLMs of different families and sizes, using the perplexity scores for each set of scenarios, and found that plausibility inference is a challenging task even for models that excel in identifying factual accuracy. Even if large models generally perform better, model architecture seems to have a more important effect of the results. with the Pythia family achieving strong scores across metrics and settings.

**Questions To Authors:**

- I was wondering if you could add an additional error analysis where you manipulate the surface form of the sentences corresponding to a set of scenarios (e.g. one way could be to try to paraphrase them via synonymy replacement, or using some syntactic transformations) and check to what extent the models' judgements are robust to such variations.
- a recent preprint you might be interested in, evaluating LLMs from a psycholinguistic point of view:
Kauf, C., Chersoni, E., Lenci, A., Fedorenko, E., & Ivanova, A. A. (2024). Comparing Plausibility Estimates in Base and Instruction-Tuned Large Language Models. arXiv preprint arXiv:2403.14859.
As an interesting point in common with your results, I see the fact that LLMs-derived log-likelihood can get you close to human-level for the distinction possible vs. impossible events, but they fall short of humans for more subtle distinctions between events with similar degrees of plausibility.

**Reasons To Accept:**

+ important and timely benchmark to evaluate LLMs capacity to discriminate the plausibility of events and scenarios. The dataset seems to be well conceived and several quality control measures have been implemented in the creation process
+ extensive evaluation with different models, prompt types and careful control of the data contamination issue, thanks to inclusion of Wikidata facts that the LLMs could not have been exposed to, given the timeframe of their training

**Reasons To Reject:**

None, for what I can see.

---

> ### Author Rebuttal · Authors · 2024-05-30
>
> Error Analysis with Surface Form Manipulation:
>
> Thank you for this insightful suggestion. We agree that examining the robustness of models' judgments through surface form manipulations, such as paraphrasing and syntactic transformations, would add significant value to our analysis. We are committed to incorporating this additional error analysis into the next version of our paper to better understand the models' robustness and variability in plausibility judgments.
>
> Related Work:
>
> We appreciate the reference to the recent preprint by Kauf et al. (2024) on evaluating LLMs from a psycholinguistic perspective. We find the parallels in their findings and our results particularly intriguing, especially the observation that LLM-derived log-likelihoods can approach human-level distinction for possible versus impossible events but fall short on subtle distinctions. We will include a discussion of this related work in our next version to provide a more comprehensive context for our findings and to highlight relevant research in this area.

---

> > ### Comment · Reviewer_9vFJ · 2024-06-05
> >
> > Thanks you for your response! I confirm my (positive) judgement of this work.

---

### Official Review · Reviewer_K7jZ · 2024-05-11

**Rating:** 7
**Confidence:** 3
**Ethics Flag:** 1

**Summary:**

This paper proposes PRobELM, a new benchmark suite for plausibility ranking evaluation for language models. \
Plausibility ranking, as the research problem the paper studies, is important. The paper also provides a systematic approach to automatically curate test cases (authors name it as "scenario"). \
The evaluation section of the paper is its strongest part; it is very extensive, covering ten models and different configurations of the testing. \
The biggest weakness for me is that there appears to be a gap between the paper's realization of plausibility and the actual definition of plausibility (check weakness section).

I am now giving a neutral score (round(5.5) = 6). I look forward to authors' responses and will raise my scores if questions clarified.

**Questions To Authors:**

Questions for the authors:

My major questions are raised in the Weakness section. Please help to address them.

Other questions:
- Can you better articulate or visualize the differences between your dataset and COPA?

- How was the hyperparameter for the 11 less plausible events determined?

- I am not very clear on the negative sampling part (Sec 3.2). Could you articulate or visualize this better? Are these samples centered around the event relations? If so, how many event relations are there in your dataset, and what is their distribution? Do different event relations undergo different processing?

- I am keen to know whether your evaluation is Markovian. P(e_2 | e_1) is different from P(e_3 | e_2, e_1). I think it is interesting to find the difference between only looking at one previous observed event and looking at all observations.

**Reasons To Accept:**

Reason to accept:
- The problem this paper tackles is important. There are many insightful intuitions in the paper, such as Bayesian inference as an approach to plausibility modeling.
- The evaluation benchmark is promising, especially concerning the temporal evolution of world facts and a language model, which is trained before (or after) a fact has become established. It is particularly useful for end users to evaluate models on recent facts to assess their reliability in a specific (new) domain and scenario.
- As mentioned in the summary, the evaluation of the paper is very comprehensive. It is beneficial to have three different evaluation settings (i.e., statement, text completion, and question answering). The consistency of models' rankings is also verified across different task settings and temporal changes.

**Reasons To Reject:**

Reason to reject:

I have some reservations about the paper's realization of plausibility in the dataset.

The definition of plausibility is "the quality of seeming likely to be true, or possible to believe" ([Cambridge Dictionary](https://dictionary.cambridge.org/dictionary/english/plausibility)).

The paper's realization in Section 3.1 (most plausible scenario) adopts a Bayesian view, considering that a future event is the most plausible given the current world state. This current realization is more suited to temporal sequence prediction rather than plausibility prediction. This is because a future event could be very surprising (e.g., Marcelo might score a goal in the 90th minute of a football match, which is very surprising (NOT plausible) given all world states in the previous 89 minutes). Can the authors substantiate how the current implementation actually models plausibility, which can also be modeling surprising events?

It is interesting to view all events $ (e_1, e_2, ..., e_n )$ in a Bayesian graph model. If I am not mistaken, the authors are now adopting a chaining view. Can the authors explain whether it is Markovian or not (i.e., future events depend only on the previous one step)? Will being Markovian or not make any difference to the empirical evaluation?

I also encourage the authors to explore the graph structure of the Bayesian graph. Is it possible that a new plausibility evaluation case can be created by $P(e_3 | e_2, e_4)$? This means that an intermediate event ($e_3$ in this case) can also be tested as long as it is not shown in the training corpus and we are certain that observed events $(e_2, e_4)$ are shown. If such a graph structure can be realized, can we make the dataset larger?

---

> ### Author Rebuttal · Authors · 2024-05-30
>
> Thanks for your review! We hope to clarify all your questions:
>
> Realization of Plausibility:
>
> The approach we take is not directly related to the surprising nature. We consider the most plausible scenarios as immediate next events that, while not recorded in the LLMs’ training data, have since become facts. Imagine traveling back to ancient times when people believed the Earth was flat; the scenario "the Earth is round" should be most plausible based on your knowledge, though it would seem very surprising to them. This shows the importance of plausibility in knowledge discovery, distinct from forecasting future events. We acknowledge that our writing in that section might have been confusing as we talked about "near future events." We will rephrase this to clarify.
>
> Markovian:
>
> Our model doesn't adopt a strictly Markovian approach. While individual timestamps in Wikidata might suggest a single-step view, the nature of world knowledge and historical events captured in LLMs is inherently non-Markovian. This broader context allows for more practical and comprehensive evaluation of plausibility, reflecting real-world applications better.
>
> Bayesian Graph Structure:
>
> Exploring the graph structure of a Bayesian model to create new evaluation cases is a promising future direction. Yet, our current focus is on scenarios that have already occurred but not in the training data.
>
> Comparison with COPA:
>
> COPA focuses on commonsense reasoning without requiring knowledge about the world. An example in COPA, "I tipped a bottle" leading to "The liquid poured out" is self-contained and doesn't challenge the model’s world knowledge. In contrast, our benchmark requires models to leverage extensive world knowledge, making the task more complex and relevant for knowledge discovery.
>
> Hyperparameter for Less Plausible Events:
>
> Regarding why we chose 10 less plausible scenarios, we aimed to make it reasonably difficult for models to score well with random guesses. We can explore with different numbers of less plausible scenarios in future versions of the paper to provide a comprehensive evaluation.
>
> Negative Sampling:
>
> Negative sampling is centered around event relations, and all 71 relations and 213 templates are detailed in the appendix D (the relations are manually selected in the QC process). This ensures a diverse and representative set of scenarios for evaluation. The full list and distribution of event relations are provided to maintain transparency and reproducibility.

---

> > ### Comment · Reviewer_K7jZ · 2024-06-06
> > **ACK your response**
> >
> > Hi authors,
> >
> > Thanks for your response. Thanks for the clarification for the first point. I would raise the score from 5.5 (neutral) to 6 as slightly positive for your paper.
> >
> > Based on your explanation, can we say that the plausible events given the current states must be seen in written corpora? If people haven't discovered the earth is round (and write them down in corpora), what will be the plausible event?

---

> > > ### Comment · Reviewer_K7jZ · 2024-06-06
> > > **Other questions**
> > >
> > > Hi authors,
> > >
> > > Sorry for my late reply!
> > >
> > > If your bandwidth allows, I have following questions:
> > >
> > > - Given there are many possibilities about the facts of the world, do you think your dataset have a good coverage about the next plausible event? For example, "the earth is round" is the case where there is only one plausible event. Will there be cases where multiple plausible events exist? What will be the coverage of your dataset on it?
> > > - Following the discussion on the multiple plausible events. What will be the distribution of the N (number of) multiple plausible events? Will it be a normal distribution?
> > > - What properties should a model possess to perform well on your task of plausibility prediction? In another word, what are we actually modelling? Are we modelling knowledge and facts? Are we modelling a capacity for sequential prediction?
> > >
> > > I hope these questions can benefit your own understanding of the topic and help AC/colleague reviewers decide.
> > >
> > > Best, K7jZ

---

> > > > ### Comment · Reviewer_K7jZ · 2024-06-06
> > > > **One final question  (thread 3/N, N=3)**
> > > >
> > > > ## Final question (thread 3/N, N=3) ##
> > > >
> > > > Hi Authors:
> > > >
> > > > I have another question. Can you help us differentiate your work against conditional QA/NLI or situated QA/NLI. For example, this paper [1] appears to adopt a pretty similar formalisation as you do in your paper (e.g. Figure 1 in this paper).
> > > >
> > > > Following the example of "earth is round", can we say your work can be seen as considering the truthfulness of a proposition conditioned on one additional fact (e.g. "ship will return to the same point after sailing for months").
> > > >
> > > > If above is true, the questions are (1) how would you differentiate your work against previous conditional / situated QA/NLI studies? (2) Do you think inferring such hidden / additional facts is the crux for solving your task? What can we expect from a model if it can perform very well on your task?
> > > >
> > > > [1] SITUATEDQA: Incorporating Extra-Linguistic Contexts into QA. EMNLP 2021. https://aclanthology.org/2021.emnlp-main.586.pdf
> > > >
> > > > Best, K7jZ

---

> > > > > ### Author Response · Authors · 2024-06-06
> > > > > **Reply to 3/3**
> > > > >
> > > > > Q: Can you help us differentiate your work against conditional QA/NLI or situated QA/NLI. For example, this paper [1] appears to adopt a pretty similar formalisation as you do in your paper (e.g. Figure 1 in this paper).
> > > > >
> > > > > A: There are at least two differences we can think of between ProbELM and SituatedQA: 1) SituatedQA is about known facts, e.g. in the example of Figure 1 in the paper all of the answers are factual, just at different points in time/geographical space. However, something can be plausible but not yet known as a fact, and this is the focus of ProbELM. 2) The (factual) answers to the SituatedQA change over time, but this is not the case in ProbELM; the shape of the earth will not change, but it is not known at the time.
> > > > >
> > > > > Q: Following the example of "earth is round", can we say your work can be seen as considering the truthfulness of a proposition conditioned on one additional fact (e.g. "ship will return to the same point after sailing for months").
> > > > >
> > > > > A: It can happen like this, but It would not always be the case. If someone did this experiment and the ship did not return to the same point (due to wind or any other reason), the earth being round would remain a plausible hypothesis.
> > > > >
> > > > > Q: If above is true, the questions are (1) how would you differentiate your work against previous conditional / situated QA/NLI studies? (2) Do you think inferring such hidden / additional facts is the crux for solving your task? What can we expect from a model if it can perform very well on your task?
> > > > >
> > > > > A: As said, the above is not always true. Having said this, a good model would perform well in our task if it is able to synthesize its knowledge in a plausible manner. As stated in the example in figure 1 of our paper, the evidence about the footballer Marcelo, if combined correctly, should make the Spanish nationality more plausible than Irish.

---

> > > > > > ### Comment · Reviewer_K7jZ · 2024-06-06
> > > > > > **Several questions are clarified**
> > > > > >
> > > > > > Hi authors:
> > > > > >
> > > > > > Thank you for your fast responses. I am sympathetic with you about your arguments about your work is about known VS unknown (in the parametric knowledge of a LM). I believe this is one key difference against prior works. I am raising my scores to 7 given such arguments.
> > > > > >
> > > > > > I still have reservations about the modelling part. Current approach is black box and it does not tell what is a LM actually modelling.  As you wrote, and I agree, that the skills is to synthesise its knowledge in a plausible manner. I believe it should be defined in a formal way -- what knowledge and in which manner (e.g. deductive or inductive reasoning?).
> > > > > >
> > > > > > Best, K7jZ

---

> > > > ### Author Response · Authors · 2024-06-06
> > > > **Reply to 2/3**
> > > >
> > > > Q: Given there are many possibilities about the facts of the world, do you think your dataset have a good coverage about the next plausible event? For example, "the earth is round" is the case where there is only one plausible event. Will there be cases where multiple plausible events exist? What will be the coverage of your dataset on it?
> > > >
> > > > A: Yes there will be cases where there are multiple plausible events. Take the Brazilian footballer Marcelo again, it is plausible that he is Spanish (he actually is, but not in training data) due to the fact he played in Spain for a number of years and the immigration policy and let’s say it is also plausible that he is Portuguese because his parents are Portuguese. In this case, our dataset would also place Portuguese near the top of the rankings because there are many Brazilians that are also Portugueses, thus the co-occurrences between these two nationalities must be frequent. Yet you are right we only evaluated the most plausible event (in this case it would be Spanish) and ignored the second plausible event (Portuguese). Extending our evaluation to multiple plausible events would be a promising future direction.
> > > >
> > > > Q: Following the discussion on the multiple plausible events. What will be the distribution of the N (number of) multiple plausible events? Will it be a normal distribution?
> > > >
> > > > A: It depends on the relation, if the relation is mutually exclusive (such as place of death), then the distribution is more concentrated whereas other relations (such as place visited) can have many plausible events. Therefore the distribution should be different from relation to relation.
> > > >
> > > > Q:What properties should a model possess to perform well on your task of plausibility prediction? In another word, what are we actually modelling? Are we modelling knowledge and facts? Are we modelling a capacity for sequential prediction?
> > > >
> > > > A: PRobELM is trying to capture the unknown/unrecorded/unthought events (not known facts at the time) that can be inferred from current world knowledge, especially in the context of knowledge discovery. PRobELM is not about forecasting the future sequential events, which would be uncertain (not guaranteed to happen).

---

> > > ### Author Response · Authors · 2024-06-06
> > > **Reply to 1/3**
> > >
> > > Thanks again for your review and additional questions
> > >
> > > Q: Based on your explanation, can we say that the plausible events given the current states must be seen in written corpora? If people haven't discovered the earth is round (and write them down in corpora), what will be the plausible event?
> > >
> > > A: No it doesn’t need to be written in corpora to be a plausible event. And such a plausible event, the earth is round, although not discovered, should be possible to infer from other knowledge such as natural phenomena that cannot be explained if the earth is not round. And better LLMs should be better able to handle this.

---

### Official Review · Reviewer_PJpF · 2024-05-14

**Rating:** 6
**Confidence:** 4
**Ethics Flag:** 1

**Summary:**

This paper introduces a Plausibility Ranking Evaluation for Language Models, a benchmark designed to assess their ability
to discern more plausible from less plausible scenarios through their parametric knowledge. In this benchmark, the LMs need to make predictions by either finishing a statement, completing the text, or completing the answer, and the authors use perplexity to rank all the candidates in the dataset to evaluate each LM's performance. Experiments are conducted on ten open source (relatively) small-size LMs to test the performance on this benchmark and show that factual accuracy (on another dataset) does not directly correlate with plausibility performance.

**Reasons To Accept:**

1. Precisely evaluating plausibility is essential in many applications, such as knowledge discovery.
2. The authors conduct experiments on many LMs.

**Reasons To Reject:**

1. First, I have doubts about the ground truth ranking in this dataset. If we use entity co-occurrences to make the ranking plausible, then basically, it's all about statistics but not reasoning. Additionally, LMs themselves are trained with loss that are highly correlated with "token" co-occurrence, and hence, I don't think it's a substantial ground truth set with this construction setting.
2. Second, it would be more appreciated if the authors could provide solutions for generating all the "ranking candidates" in real applications. For example, if conducting knowledge discovery, how would this plausibility benchmark help us get and rank all plausible candidates? If we still rely on LMs to generate the results and use the perplexity to rank, what would be the difference between it and beam search?
3. I think deeper analysis should be conducted, especially on the observations the paper found -- is there any reason and insights on "higher performance on factual accuracy do not necessarily excel in plausibility" ; Any deeper analysis on the drop of 2017 performance drop?

---

> ### Author Rebuttal · Authors · 2024-05-30
>
> Thanks for the review.
>
> Ground Truth Ranking:
>
> We use information from Wikidata added after the training cut-off of the LLMs. This ensures our benchmark evaluates their ability to infer new world knowledge beyond co-occurring words. Our analysis shows that the most frequently co-occurring entity is not always the most plausible entity (>~83% of the cases), illustrating that plausibility often requires reasoning beyond mere co-occurrence.
>
> Generating Ranking Candidates in Real Applications:
>
> Our benchmark can assess the plausibility of selected outcomes but not an exhaustive list. Extending our benchmark to rank a full list of outcomes is a promising future direction and we plan to achieve this with human annotation. This will enhance our benchmark in real-world applications, such as knowledge discovery, where identifying the most plausible hypotheses from a broader set is crucial.
>
> Difference from Beam Search:
>
> Beam search could indeed be used to obtain more plausible hypotheses, and its use is orthogonal to evaluation with our benchmark. However, maximum a posteriori estimates are not always the best way to decode a model, which is why minimum Bayes risk decoding is popular in fields like machine translation. Our method focuses on evaluating the plausibility of scenarios using perplexity, providing a different perspective than beam search.
>
> Deeper Analysis on Key Observations:
>
> Our finding that higher performance on factual accuracy does not necessarily correlate with better plausibility performance can be attributed to the distinct nature of these tasks. Factual accuracy tests a model's ability to recall and verify known facts, while plausibility requires the model to infer and rank scenarios that might not be explicitly present in its training data but are logical extensions of its knowledge. For example, Marcelo being Brazilian is factual, but Marcelo being Spanish is plausible due to his history and Spanish naturalization laws.
>
> The drop in performance with the 2017 dataset is due to the specific characteristics of the data used. The "occupation" relation disproportionately influenced the performance. When “occupation” involves uncommon terms like "valets de chambre," the perplexity is very low across all models, making it the top choice regardless of the subject entity. This occurs because such specific terms might have been less frequently represented in the training data, causing models to assign lower perplexity to these terms when they appear.

---

> > ### Comment · Reviewer_PJpF · 2024-06-06
> > **Response from Reviewer**
> >
> > Thank you for the explanation. I get most of the rebuttal ideas. For the "Ground Truth Ranking" concern, I would expect a more thorough analysis to be presented in the later version of the paper.
> >
> > Another concern I have for this paper is that this paper is built for each LM and uses the corresponding cutoff time to generate the data. How do you expect this benchmark to be long-lasting for future research and development on LM?
> >
> > If this additional concern can be answered, I'll further raise my evaluation.

---

> > > ### Author Response · Authors · 2024-06-06
> > > **Additional rebuttal**
> > >
> > > Ground Truth Ranking:
> > >
> > > We will report more detailed statistics in the next version.
> > >
> > > Future LLMs:
> > >
> > > Benchmark is built for each LLM: Your concern is reasonable, however our benchmark can be reused as long as the training data of the LLM is not more recent than the evaluation time stamps of the benchmark. See in Table 1, that OLMO and LLama are evaluated on the same version of the benchmark, since the timestamps considered are more recent than the training data. Furthermore, we can always remove evaluation scenarios as more recent training data is used. Finally, the benchmark construction process is fully automated, so even if we needed to create new scenarios for later timestamps than the ones we considered in this version, it would only require running our code on more recent versions of wikidata.

---

> > > > ### Comment · Reviewer_PJpF · 2024-06-06
> > > > **Response from Reviewer**
> > > >
> > > > Hi Authors,
> > > >
> > > > Thank you for your explanation. However, from the paper, especially Section 3.3, I don't see the dataset creation is fully automated. Can you elaborate more about this?
> > > >
> > > > Thank you.

---

> > > > > ### Author Response · Authors · 2024-06-06
> > > > > **Additional rebuttal**
> > > > >
> > > > > Thanks again for your review and additional question.
> > > > >
> > > > > Sorry for the confusion. In the response we meant that: using our 71 relations, extending our dataset to a future version of Wikidata for future models is fully automated, as the quality control (sec 3.3) is relation specific not timestamp specific. Given that we have done the quality control, the rest of the data construction is fully automated therefore extending is also fully automated.

---

### Decision · Program_Chairs · 2024-07-10

**Decision:**

Accept

**Comment:**

This paper proposes a benchmark construction method, and specific instantiations, for assessing the ability of language models to identify more or less plausible scenarios. The reviewers were initially mildly positive and after the rebuttal and discussion phase became more strongly positive.

The reviewers found the paper's contribution to evaluating plausibility assessment to be important and timely. They appreciated the extensive experiments and the quality controls implemented in the benchmark.

The discussion period raised two important points:
1. how relevant this benchmark is for future models, given that it's built with snapshots of Wikipedia edits. The authors provided sufficient answers here. I would strongly suggest adding a thoughtful discussion of this in the paper, as well as a concrete procedure in the released code base.
2. how to define plausibility and what the benchmark is testing for vs other prior datasets (discussion with Reviewer K7jZ). The discussion was insightful and again, the authors should integrate it into the paper.

Given the discussion and trusting the authors edit the paper accordingly, I recommend acceptance and believe this is a useful contribution to the community.